

# A knowledge graph embedding model based attention mechanism for enhanced node information integration

Ying Liu[1,2], Peng Wang[1,3], Di Yang[1] and Ningjia Qiu[1]

[1] School of Computer Science and Technology, Changchun University of Science and Technology, Changchun, China
[2] School of Computer Science, Tonghua Normal University, Tonghua, China
[3] Changchun University of Science and Technology Chongqing Research Institute, Chongqing, China

## ABSTRACT

The purpose of knowledge embedding is to extract entities and relations from the knowledge graph into low-dimensional dense vectors, in order to be applied to downstream tasks, such as connection prediction and intelligent classification. Existing knowledge embedding methods still have many limitations, such as the contradiction between the vast amount of data and limited computing power, and the challenge of effectively representing rare entities. This article proposed a knowledge embedding learning model, which incorporates a graph attention mechanism to integrate key node information. It can effectively aggregate key information from the global graph structure, shield redundant information, and represent rare nodes in the knowledge base independently of its own structure. We introduce a relation update layer to further update the relation based on the results of entity training. The experiment shows that our method matches or surpasses the performance of other baseline models in link prediction on the FB15K-237 dataset. The metric Hits@1 has increased by 10.9% compared to the second-ranked baseline model. In addition, we conducted further analysis on rare nodes with fewer neighborhoods, confirming that our model can embed rare nodes more accurately than the baseline models.

# INTRODUCTION

In order to improve the quality and user experience of search engines, Google first proposed the concept of the knowledge graph (*Singhal, 2012*), credited to the superior performance of knowledge graphs in intelligent services, more and more companies are launching their own knowledge graphs products. For example, the famous public knowledge graphs Freebase (*Bollacker et al., 2008*), Yago (*Suchanek, Kasneci & Weikum, 2007*), Wikidata (*Vrandecic & Krtoetzsch, 2014*) have stored a vast amount of knowledge. Currently, knowledge graph has been used in a variety of many application fields, including intelligent search, recommender systems (*Palumbo, Rizzo & Troncy, 2017*; *Wang et al., 2018*), question answering (*Bordes, Weston & Usunier, 2014*; *Bordes, Chopra & Weston, 2014*), decision-making, *etc.* The knowledge graph takes the graph structure as the

Corresponding author
Peng Wang, wangpeng@cust.edu.cn

knowledge carrier, the nodes within the graph represent concrete facts and abstract concepts, and the edges represent the relationships between these objects. The nodes in the knowledge graph and their binary relations are stored in the form of triples to describe objective facts in the real world. The knowledge representation learning of knowledge graph is the basis of the construction and application. It aims to accurately and efficiently express the implicit semantics in entities and relations with low dimensional and dense vectors, in order that computers can use knowledge for calculation, reasoning and other processes to better serve downstream applications.

A variety of classical knowledge representation models have been extensively investigated and employed, including the translation model TransE (*Bordes et al., 2013*) and its enhanced version, as well as semantic-based models such as RESCAL (*Nickel, Tresp & Kriegel, 2011*) and DistMult (*Yang et al., 2014*). These models have attracted significant attention and have been extensively applied in various domains. Notably, the knowledge representation acquired through these conventional models of single triplets has demonstrated its effectiveness in tasks such as entity classification and link prediction. Many research efforts have been dedicated to exploring and deploying these models, confirming their value in practical applications.

However, a knowledge graph is essentially a complex semantic network. To accurately represent knowledge, we should not only rely on the relation between a single triplet, but also take into account the graph of related multi-source information and its internal structure. Graph structures are commonly used as information carriers in complex systems. The embedding representation of learning graphs, which converts information into an optimal vector representation, is one of the methods for solving graph representation information. Topological neighborhood aggregation-based graph embedding models learn the vector representation of a node by aggregating the information from the neighboring domains of the central node. This is typically done using an information aggregation function operation to combine the information from a cluster of neighborhoods and generate a vector representation of, such as graph neural networks (*Donghan et al., 2021*; *Palash & Emilio, 2018*). Compared to other networks, graph neural networks are able to aggregate entity neighborhoods in non-Euclidean space and have superior representation ability. The famous graph convolution network (GCN) (*Kipf & Welling, 2016*) model applies the concept of convolution to the non-Euclidean space. It aggregates neighborhood information through message passing methods to enhance the knowledge representation of central nodes, while the graph attention network (GAT) (*Velickovic et al., 2017*) assigns different attention weight values to different neighborhoods, in order to more effectively aggregate their information. However, there are still many defects in their representation methods. For example, the GCN model cannot accept the addition of new nodes and has insufficient representation ability for directed graphs, while GAT is not effective in aggregating multi-level neighborhood information and is sensitive to initialization parameters. Additionally, these methods lack techniques for expressing rare nodes.

Our contributions can be summarized as follows:

1. We have put forward a novel model named KATKG that enhances the method of aggregating key node information, leading to a more effective representation of uncommon

nodes. Furthermore, this model facilitates the establishment of connections between nodes throughout the entire graph, even for those with few links. This enhances the participation and connectivity of uncommon nodes within the graph as a whole.

2. We integrate attention mechanisms into the knowledge representation of heterogeneous knowledge graphs to enhance the embedding representation of entities in the knowledge graph. Additionally, we propose a relationship update layer to embed relationships.

3. We instantiate our model and evaluate its link prediction results on FB15k-237 and WN18RR show that our model and confirm that its embedded representation has significantly improved the performance on rare nodes.

## RELATED WORK

The classical translation model TransE (*Bordes et al., 2013*) proposes that in the vector space, the distance from the head vector to the tail vector approximately represents the relationship between them. Although TransE is simple and effective, it is still unable to fit such complex relations as 1 to N, N to 1, N to N. TransR (*Lin et al., 2015*), a variant of TransE, maps the relation to a new relation specific vector space, and TransH (*Wang et al., 2014*) introduces a relation-specific hyperplane, while other improved models based on TransE, such as TransD (*Ji et al., 2015*), TransM (*Fan et al., 2014*), and TransF (*Feng et al., 2016*), have solved the problem that TransE cannot handle multiple relations to a certain extent. RotatE (*Sun et al., 2019*) provides a brand-new embedding idea, it uses the principle of Euler rotation, regards the rotation vector between entities as a relation and maps both the entity and relation to the complex space, and obtains a better connection prediction compared to TransE. However, the TransE model and its derived models do not have a mechanism for directly aggregating neighborhood information. It primarily focuses on the transformational relationship between entities and relations in triplets, capturing their semantic associations by learning the vector representation of entities and relationships.

The models based on semantic matching and translation-based approaches have limitations in that they only consider information from the triplets, disregarding the structural information of the graph. Semantic matching energy (SME) (*Bordes et al., 2012*) is an early semantic matching model, which uses neural networks to accomplish semantic matching. In the hidden layer, the score vector obtained by combining the head vector and the relationship vector is then combined with the score vector obtained by the tail entity to calculate the matching score. The neural tensor networks (NTN) (*Socher et al., 2013*) model first represents the entity as an embedding vector, maps the relationship of the model to a tensor, and then maps it to a nonlinear hidden layer. Although the model has strong representation ability, it has too many parameters to make the model unable to be applied to largescale knowledge graphs. RESCAL and its extended model group are relatively classical semantic matching models, it uses vectors to represent relations and matrices to represent semantic relations among all entities. Distmult has made improvements on the RESCAL model by constraining the relationship matrix to a diagonal matrix. This method significantly reduces the time complexity, but it also causes Distmult to be unable

to fit asymmetric relations and reduces the accuracy of the model. ComplEx (*Trouillon et al., 2016*) introduced complex numbers on the basis of Distmult and improved it to fit asymmetric relations, and expressed different semantic information through different head and tail sequences of triplets.

The semantic matching-based models and the translation-based models only consider the information from triples but ignore the structural information of the graph. For example, in the knowledge graph, the representation of information about an entity with human attributes will be affected by the nationality, education level, family members, and friends of the entities around it. Therefore, for the representation of entities and relationships, in addition to learning the semantic information of triplets, we should also combine the graph structure information to more fully express knowledge.

Graph neural networks (GNN) (*Zhou et al., 2020*) consider the topological structure between nodes, it has a strong ability to express non-Euclidean space node information based on graph structure, which makes it widely used in social networks, intelligent recommendation, knowledge reasoning and other fields. *Kipf & Welling (2016)* proposed the graph convolutional network (GCN), which integrates the idea of convolution, extracts the structural features of the graph, aggregates neighborhood information through the weight matrix and obtains a feature-embedded representation of the entity through hierarchical propagation and aggregation. which integrates the idea of convolution, extracts the structural features of the graph, aggregates the neighborhood information through the weight matrix, and obtains the feature embedded representation of the entity through hierarchical propagation and aggregation. GCN can effectively capture global information, thus representing the node characteristics of the graph well. It has achieved remarkable results in the semi supervised node classification experiments. But GCN can only assign fixed weights to neighborhoods, and all nodes need to be retrained in each iteration to get the node embedding on GCN, it is impossible to quickly get the embedding of new nodes.

The GraphSAGE (*Hamilton, Ying & Leskovec, 2017*) does not learn fixed embeddings for each node, but trains a group of aggregator functions that can aggregate local neighbor feature information, aggregates a fixed size of neighborhood information through random K-step walks and only selects some neighborhoods for aggregation, so that it can quickly train the representation information for new nodes. *Velickovic et al. (2017)* proposed GAT; this model does not rely on the structure of the graph itself, but aggregates neighborhoods nodes information through a self-attention mechanism. It calculates the attention weight values between nodes to represent the importance of other nodes to the central node, and realizes the adaptive matching of weights for different neighborhoods, so that the model performance and stronger robustness.

When applying graph neural network models to knowledge graphs, several issues may arise, such as the inadequate representation of relation information. There are some problems in applying graph neural network models to knowledge graphs, such as the inability to effectively represent the relation information. Many researchers have observed this and made enhancements, allowing graph neural networks to be applied to knowledge graphs with more intricate structures. Relational graph convolutional

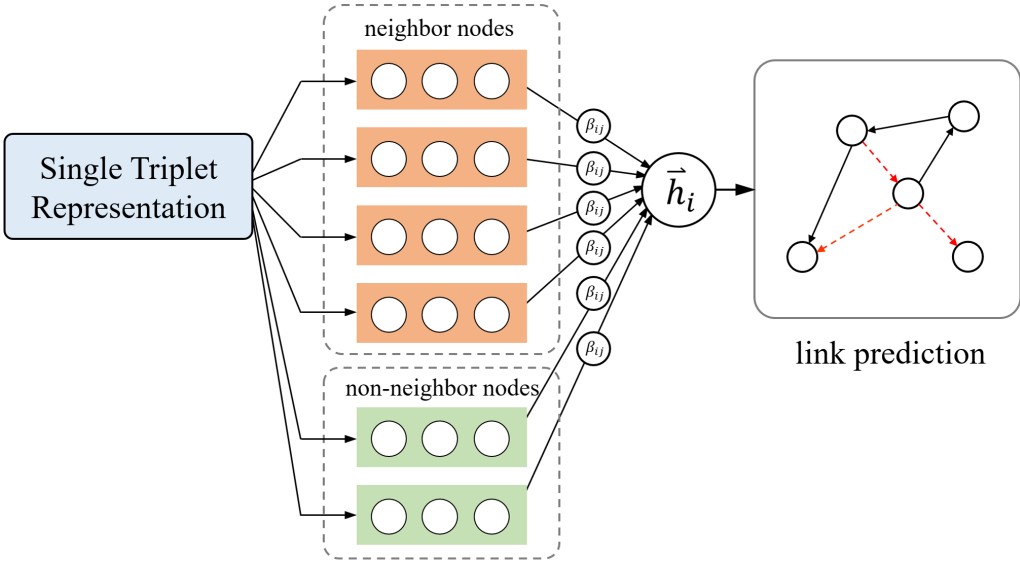

**Figure 1  Overview of our model.**

networks (RGCN) (*Schlichtkrull et al., 2018*) is applied to relational data modeling on the basis of GCN, so that makes the model can represent the information implied in the relation. RGCN aggregates neighborhoods according to the different relations and performs corresponding conversion according to the different types of edge. After regularization, weighted summation, and activation processing, the final embedding is obtained. TransGCN (*Cai et al., 2019*) integrates the translation hypothesis in TransE with GCN, and the rotation hypothesis in Rotate with GCN. It sets different conversion operators based on the direction of the relation. TransGCN completes the relation on the basis of traditional GCN, uses fewer parameters than RGCN, and achieving better experimental results. *Zhang et al. (2020)* proposed relation graph neural network (RGHAT). This model includes a relation-level attention layer and an entity-level attention layer. In the relation-level attention layer, the central node aggregates different relations based on different attention weights. In the entity-level attention layer, the central node aggregates different neighboring entities based on different attention weights within the same relation. The hierarchical attention mechanism improves the interpretability of the model.

## METHODOLOGY

The technical details of our model are provided in this section; the overview is shown in Fig. 1.

### Symbols and definitions

The knowledge graph is essentially a directed graph, which is composed of nodes representing entities and edges representing relationships. The edges in a knowledge graph are directed, and it contain semantic information about the relationships between nodes. We use the symbol G to represent the entire knowledge graph, denoted as

$G = \{(e_h, r, e_t)\} \subseteq E * R * E$, where E represents the set of all entities in the knowledge graph and R represents the set of relationships. Each triplet in the knowledge graph is represented as $(e_h, r, e_t)$, where $e_h$ and $e_t$ represent the head and tail entity elements, $r$ represents the relationship between $e_h$ and $e_t$, and embedded $e_h$, $r$, $e_t$ into m-dimensional vector space.

## KATKG

We refer to the model proposed in this article as KATKG, which incorporates the neighborhood aggregation method of GAT for reference and the translation hypothesis of TransE. It can represent entities and relationships in heterogeneous knowledge graphs through vectorization. Our model includes three layers: the single triplet representation layer, the key information aggregation layer, and the relation update layer.

### Single triplet representation layer

We use the single triplet representation layer to represent the initialization vector, there are two main reasons for this. First, our model is based on the graph attention network to improve the knowledge representation learning task. Although the graph attention network can constantly update and integrate the information from other nodes into its own knowledge representation, it does not pay enough attention to the interactive information contained within the triplet itself, and focuses more on the information representation of the graph structure. Secondly, GAT is sensitive to parameter initialization, therefore, we believe that we can achieve improved results by initially training the mutual semantic information between triples to acquire the initialization vector representation of entities and relationships, and then aggregating the domain information through the graph attention mechanism.

TransE is a translation-based model, which proposes that the head entity and tail entity of a triple $(e_h, r, e_t)$ can be mapped into vector space, and the relationship between them can be expressed as $e_h + r \approx e_t$. We select the TransE model method to fit information into a single triplet and pretrain entities and relationships to achieve better convergence and enhance the stability of the model. The TransE model considers that the fact expressed by a correct triplet satisfies $e_h + r \approx e_t$, $d(e_h + r, e_t)$ should reach a minimum value as far as possible. On the contrary, for a corrupted triplet, $d(e_h + r, e_t)$ should be as large as possible, where the function $d$ represents the distance between $e_t$ and $e_h + r$ mapped to the vector space, the loss function is as follows:

$$L = \sum_{(e_h, r, e_t) \in S_{(e'_h, r, e'_t)}} \sum [\gamma + d(e_h + r, e_t) - d(e'_h + r, e'_t)]_+ \tag{1}$$

where $[x]_+$ denotes the positive part of $x$, and $\gamma$ is a margin hyperparameter. In our experiment, the distance formula adopts L2 regularization constraints. The construction method of negative example triplets $(e'_h + r, e'_t)$ is to replace one of the head or tail entities of the correct triplet with another entity. The set $S'_{(e_h, r, e_t)}$ of the negative example triplet is as follows:

$$S'_{(e_h, r, e_t)} = \{(e'_h, r, e_t) | e'_h \in E\} \cup \{(e_h, r, e'_t) | e'_t \in E\}. \tag{2}$$

By minimizing the loss function, minimizing the distance of positive examples and maximizing the distance of negative examples, to achieve the goal of training the knowledge representation of entities and relations.

### Key information aggregation layer

We use $h = \left\{ \vec{h}_1, \vec{h}_2, \ldots, \vec{h}_N \right\}, \vec{h}_i \in \mathbb{R}^m$ to represent the vector representation of entities obtained through the TransE model, where $\vec{h}_i$ represents the feature vector representation of entities, the $m$ is the dimension of vector $\vec{h}_i$ and the $N$ represents the number of nodes. Before aggregating the information from neighborhoods, we need to perform linear transformations in order to enable input features to obtain more information and enhance their expression ability. The entity vector $h$ is linearly transformed by the feature matrix $W \in \mathbb{R}^{m' \times m}$ to obtain a new node feature vector $h\prime = \left\{ \vec{h}\prime_1, \vec{h}\prime_2, \ldots, \vec{h}\prime_N \right\}, \vec{h}\prime_i \in \mathbb{R}^{m'}$, where $\vec{h}\prime_i$ represents the processed node vector representation and $m\prime$ is the dimension of $\vec{h}\prime_i$.

We use the self-attention calculation method without considering the graph structure to determine the weight value of the interaction between each pair of nodes. The calculation method is to use the concatenation operation to concatenate $\vec{h}\prime_i$ and $\vec{h}\prime_j$ into 2m-dimensional vectors, then calculates dot product with transposition of vector $\vec{a}$, to construct a mutual self-attention matrix $X \in \mathbb{R}^{N \times N}$, which $\vec{a}$ is a learning weight vector parameterized.

$$X = \vec{a}^T \left( W\vec{h}'_i || W\vec{h}'_j \right). \tag{3}$$

Then activated by LeakyReLU function to obtain $\alpha_{ij}$, that is the attention weight value of node $j$ to node $i$. The self-attention is the interaction weight between all pairs of nodes in the global without considering the graph structure, this attention weight represents the importance of every two nodes.

$$\alpha_{ij} = \text{LeakyReLU}(X). \tag{4}$$

The LeakyReLU function is a neural network unit activation function that can solve both the vanishing gradients and dying ReLU problems. The mathematical expression for the LeakyReLU function is as follows:

$$y = \max(0, x) + leak \times \min(0, x). \tag{5}$$

Where the parameter *leak* is a small constant that refers to the degree or rate of leakage, which we have chosen as 0.01 in our experiments.

The training based on triplets and the training based on graph structure largely depend on the connectivity between nodes, knowledge graphs usually contain many rare nodes with low connectivity or even no links, which are difficult to train. Generally, the knowledge graph is a complex network, and the number of neighborhoods for each entity varies enormously. Some entities have hundreds of neighborhoods, while some entities have only a few or even none at all. Entities with more neighborhoods can embed richer graph structure information, while entities with fewer neighborhoods cannot rely on graph structure information to enrich their own semantics. Therefore, we supplement nodes

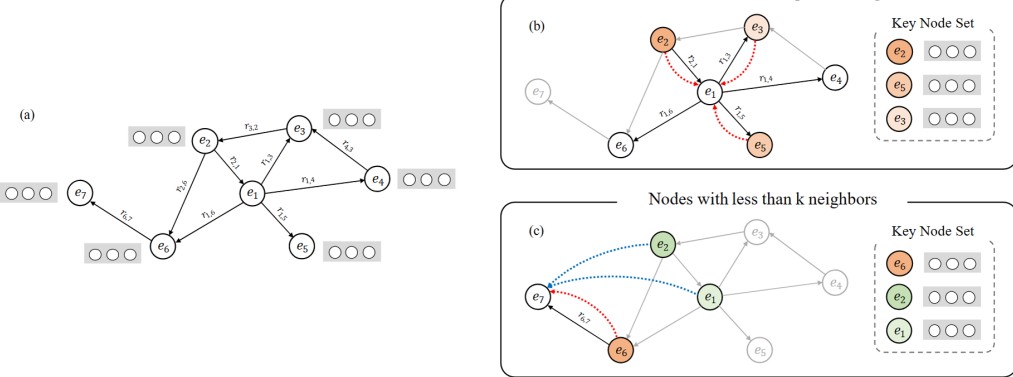

**Figure 2 Visual description of key node set construction method.** (A) A micro knowledge graph example. (B) Aggregation method for nodes with more than or equal to k (k = 3) neighbors, e1 is the central node. (C) Aggregation method for nodes with less than k (k = 3) neighbors, e7 is the central node.

set with potential relationships for nodes with fewer than k neighborhoods, even if there are no connected edges between nodes, the weight value of self-attention can be used to approximately represent the impact of the whole graph on rare nodes, which strengthens the participation of rare nodes and the connection ability with the whole graph, and improves the representation ability of the model for rare nodes.

In the normalization processing of the neighborhood information aggregation layer, we consider the weight values of $k$ key nodes with most import information for the central node, which $k$ is a controllable hyperparameter. We consider two cases here: For the nodes with more than or equal to $k$ neighborhoods in the graph, only the $k$ neighborhoods with the highest interaction weight need to be filtered and calculated for the neighborhoods. For the nodes with fewer than $k$ neighborhoods, we divide the set into two parts, the first part is all the neighborhoods of the node, for the remaining insufficient part, we filter the most important key nodes from $\alpha_{ij}$ to supplement the set to $k$ neighborhoods, these two parts are combined to form a key node set of rare nodes, as shown in Fig. 2.

Our specific method for aggregating node information is as follows: (i) If the current node $i$ has $c$ neighborhoods ($c \geq k$), our method of forming the key node set K is to select the first $k$ nodes with high attention weight values from the neighborhoods of node $i$. (ii) If the current node $i$ has $c$ neighborhoods ($c < k$), our method of forming the key node set K is including all neighborhoods, and then calculate the global node's attention weight value for the current node $i$. Merge the highest $k - c$ nodes into the set K, if there are neighborhoods for the current node $i$, skip it and continue searching downwards.

After obtained the key node set K, we use the *softmax* function to normalize and calculate the attention weight of all neighborhoods, so that the coefficients can be easily compared between different nodes:

$$\beta_{ij} = softmax_j\left(e_{ij}\right) = \frac{\exp(\alpha_{ij})}{\sum_{n \in K} \exp(\alpha_{in})}. \tag{6}$$

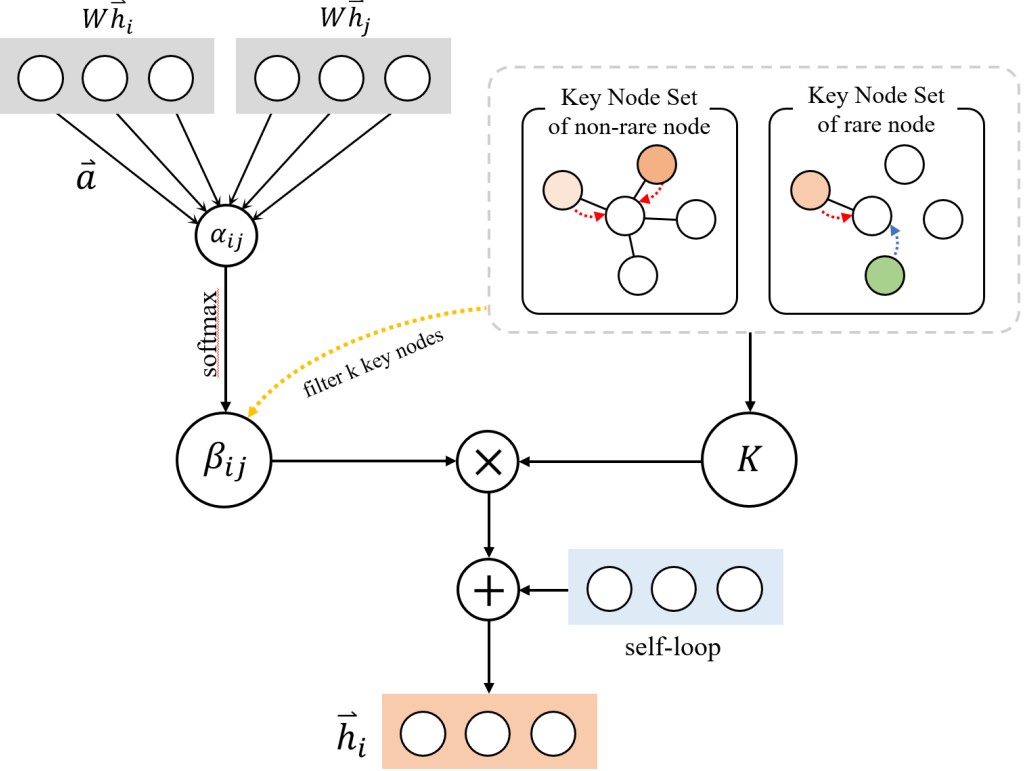

**Figure 3** Structure of key information aggregation layer.

$\beta_{ij}$ is the normalized attention coefficients of the central node and its key nodes. Finally, we aggregate the information of $k$ key nodes, introduce the weight matrix $W$, and obtain the hidden representation of nodes through linear transformation:

$$\vec{h}_i = \sigma\left(\sum_{j \in K} \beta_{ij} W \vec{h}_j\right) + W \vec{h}_i. \tag{7}$$

We adopt the method of adding the vector representation of the central node after the weight matrix $W$ transformation to strengthen the semantics of the central node, $\vec{h}_i$ is the final vector which contains the information of the central node and the aggregated neighborhood information. The KATKG model filters $k$ key nodes for aggregation instead of aggregating all neighborhoods, it can limit the aggregation of redundant information in the case of too many neighborhoods and improves the aggregation effect for rare nodes. Figure 3 shows the structure of the key information layer.

### Relation update layer

After key information aggregation layer, the hidden representation of entities has been trained, but the hidden representation of the relation still remains as the vector representation before training. Therefore, we introduce a relation update layer to update the relation embedding vector. According to the translation hypothesis proposed by

**Table 1   Basic statistics of WN18RR and FB15k-237.**

| Dataset | WN18RR | FB15k-237 |
|---|---|---|
| Entities | 40,943 | 14,541 |
| Relations | 11 | 237 |
| Training | 86,835 | 272,115 |
| Validation | 3,034 | 17,535 |
| Test | 3,134 | 20,466 |

*Bordes et al. (2013)*, we update the embedded representation of the relation by traversing the difference of entity vectors between different triplets of the same relation.

$$r_i' = \frac{1}{n}\sum_{n \in R_i}(e_{tn} - e_{hn}) + r_i. \tag{8}$$

where $R_i$ is a set of all triplets that include relation $i$, $r_i$ is the embedding representation of the relation obtained from the single triplet representation layer, and $r_i'$ is the embedding representation of the finally relation.

# EXPERIMENT

## Datasets

We use the WN18RR (*Dettmers et al., 2018*) and FB15k-237 (*Toutanova & Chen, 2015*) datasets in the link prediction experiment to evaluate the performance of KATKG. WN18 is a subset derived from WordNet, while FB15k is a well-established subset in the field of knowledge graph learning, specifically Freebase. However, both WN18 and FB15k are affected by inverse relations. To address this issue, we utilize refined subsets of the original datasets, specifically WN18RR and FB15k-237, in our experiment. These subsets are smaller in size but offer more precision by eliminating the impact of inverse relations. For both the WN18RR and FB15k-237 datasets, we divide them into training, validation, and testing sets. Detailed statistics regarding these datasets are presented in Table 1.

## Experiment setup
### Evaluation metrics

To evaluate the effectiveness of our model, we conducted link prediction experiments on the WN18RR and FB15k-237 datasets. In these experiments, we construct corrupted triplets by replacing either the head or tail entity of each correct triplet. By minimizing the loss function, we train the vector representations of entities and relations. To assess the performance of our model, we utilize several evaluation metrics. We use the Hits@k (Hit@1, Hit@3, Hit@10), mean ranking MR and mean reciprocal rank MRR to evaluate the model. MR represents the average ranking of all correct triples, so a lower value indicates better model performance. MRR represents the sum of the reciprocal rankings of all correct triplets, so a higher value indicates better performance of the model; Hits@k indicates that the correct triplet ranking is less than the average proportion of k, as the same of MRR, a higher value indicates better performance of the model.

**Table 2** Neighbor Statistics of Training Set.

|  | WN18RR | FB15k-237 |
|---|---|---|
| Entity number | 40943 | 14541 |
| Number of entities without neighbors | 384 | 36 |
| Number of entities with neighbors less than five | 29784 | 1318 |
| Maximum number of neighbors | 482 | 7614 |
| Number of neighbors owned by the most nodes | 2 | 11 |
| Average number of neighbors | 4.24 | 37.42 |

**Table 3** The ranges and values of key experimental parameters.

| Parameters | Range of parameter values | | WN18RR | FB15k-237 |
|---|---|---|---|---|
| $m$ | 50,100,200 | | 50 | 50 |
| $L$ | 1,2 | | 2 | 2 |
| $m_d$ | 100,200,300 | | 100 | 200 |
| $\lambda$ | 0.01,0.005,0.001 | | 0.005 | 0.01 |
| $\gamma$ | 1,2 | | 2 | 1 |
| $k$ | 3, 5, 10 (in WN18RR) | 5, 10, 20 (in FB15k-237) | 3 | 10 |

### Implementation details

In the training process, we use the Adam optimizer and select the following hyperparameters: entity and relation embedding dimension $m$ among $50, 100, 200$, layer depth $L$ among $1, 2$, hidden layer dimension among $m_d$ $100, 200, 300$, fixed learning rate $\lambda$ among $0.01, 0.005, 0.001$, and margin $\gamma$ among $1, 2$. In the experiment, we need to select key $k$ neighborhoods to aggregate information, so we perform a statistical analysis of the number of neighborhoods of each entity in each set. The details are shown in Table 2.

In the WN18RR dataset, the number of nodes with 2 neighborhoods is the largest, the average number of neighborhoods of a node is 4.24, therefore, we chose $k$ value is among $3, 5, 10$ in the WN18RR dataset. In the FB15k-237 dataset, the average number of neighborhoods for nodes is 37.42, and most number of nodes have 11 edges, therefore, the $k$ value is among $5, 10, 20$ on the FB15k-237 dataset.

We select the above parameters for training, and retain the best parameters when MRR obtains the best performance on each verification set. For WN18RR, the optimal parameters are entity and relation embedding dimension 50, layer depth 2, hidden layer dimension 100, learning rate 0.005, margin $\gamma = 2$, and number of neighborhoods $k = 3$. For FB15k-237, the optimal parameters are entity and relation embedding dimension $m = 50$, layer depth $L = 2$, hidden layer dimension $m_d = 200$, learning rate $\lambda = 0.01$, margin $\gamma = 1$, and number of neighborhoods $k = 10$, the ranges and values of key experimental parameters are shown in Table 3 for clarity.

**Table 4  Prediction results on the WN18RR and FB15k-237 datasets.**

| | WN18RR | | | | | FB15k-237 | | | | |
|---|---|---|---|---|---|---|---|---|---|---|
| | MRR | MR | Hits@ | | | MRR | MR | Hits@ | | |
| | | | 1 | 3 | 10 | | | 1 | 3 | 10 |
| DistMult[a] | 0.43 | 5110 | 0.39 | 0.44 | 0.49 | 0.241 | 254 | 0.155 | 0.263 | 0.419 |
| TransE[a] | 0.226 | 3384 | - | – | 0.501 | 0.294 | 357 | - | – | 0.501 |
| ComplEx[a] | 0.44 | 5261 | 0.41 | 0.46 | 0.51 | 0.247 | 339 | 0.158 | 0.275 | 0.428 |
| ConvE[a] | 0.43 | 4187 | 0.40 | 0.44 | 0.52 | 0.325 | 244 | 0.237 | 0.356 | 0.501 |
| ConvKB[b] | 0.265 | 1295 | 0.058 | 0.445 | 0.558 | 0.289 | 216 | 0.198 | 0.324 | 0.471 |
| R-GCN[b] | 0.123 | 6700 | 0.08 | 0.137 | 0.207 | 0.164 | 600 | 0.10 | 0.181 | 0.30 |
| KATKG | 0.296 | 1885 | 0.277 | 0.448 | 0.567 | 0.32 | 212 | 0.263 | 0.378 | 0.553 |

**Notes.**
[a]The results are taken from *Sun et al. (2019)*.
[b]The results are taken from *Nathani et al. (2019)*.

# RESULT

## *Main result*

We use the Hits@1, Hits@3, Hits@10, MR and MRR as evaluation indicators. Table 4 shows the results of our model in WN18RR and FB15k-237 datasets, as well as the comparison results with some baseline models.

According to the findings presented in Table 4, our model demonstrates competitive performance on the WN18RR dataset, particularly in the Hits@10 metric. Nevertheless, our model does not exhibit satisfactory performance on other metrics. One possible explanation for this is that the WN18RR dataset has a lower number of relationships compared to other datasets, which leads to sparser connections between nodes. On the FB15K-237 dataset, our model outperforms the baseline model in all MR metrics except for the MRR metric. Although our model ranks behind ConvE in terms of the MRR metric, it demonstrates commendable performance in the Hits@1, Hits@3, and Hits@10 metrics. In summary, our model exhibits noteworthy performance on both the WN18RR and FB15K-237 datasets, showcasing its superiority compared to the baseline model. These results provide evidence of the effectiveness of our model and emphasize the value of leveraging attention mechanisms to acquire valuable information from local neighborhoods and key nodes.

## *Result of rare node embedding*

In the training set, rare nodes with fewer connections in the graph often pose challenges in accurately representing them. This is primarily due to the inadequate training received by nodes with limited neighborhoods. To overcome this limitation, our model integrates information from the triplets and graph structure using a global attention mechanism. This approach enhances the effectiveness of embedding rare nodes and improves the overall connectivity of the graph.

According to statistics, the FB15k-237 dataset contains 36 isolated nodes, 365 nodes with only one neighborhood, and 1318 nodes with less than five neighborhoods, accounting for 9% of the total dataset. To verify the accuracy of our model in embedding rare nodes, we screen out the nodes without neighborhoods in the training set of FB15k-237 in the

**Table 5** Ranking of rare node embedding results.

| Entity name | TransE | KATKG |
|---|---|---|
| /m/05hyf | 10747 | **4263** |
| /m/0qb7t | 12931 | **7536** |
| /m/027qb1 | 12895 | 13821 |
| /m/026y05 | 10942 | **8027** |
| /m/03tp4 | 6860 | **5644** |
| /m/0lyb_ | 13915 | **12218** |
| /m/047vnfs | 3015 | **1052** |
| /m/022qqh | 3104 | **1051** |
| /m/027yjnv | 7704 | 12008 |
| /m/024030 | 11893 | **5577** |

**Notes.**
Values marked in bold indicate that our model achieves a higher ranking than TransE.

experiment, and ensured that node existed in the triplet of the test set. All nodes in the test set are used to replace the triplet heads and tails of rare nodes in order to create a corrupted triplet set. We randomly selected 10 rare entities which filled the requirements for analysis, and the final ranking of the correct triplet is shown in Table 5.

The result shows that eight of ten eligible entities are better than the results of TransE. That indicates our model can improve the fitting degree of rare nodes, and proves that the method of aggregating global key nodes is meaningful.

## CONCLUSIONS

In this article, we propose a knowledge embedding model that incorporates the attention mechanism. Our model effectively aggregates the information from k key nodes to complement node information while mitigating the impact of unimportant nodes. Furthermore, our model utilizes the global attention mechanism to enhance the embedding representation of rare nodes and facilitate their meaningful integration into the overall graph structure. We have experimentally confirmed the reliability of our model. Furthermore, we extracted rare nodes from the FB15K-237 dataset to assess whether our model improves their representation. Our model performs well on the FB15K-237 dataset, with most parameters outperforming the classical limit model. However, our model underperforms on the WN18RR dataset, which indicates that while our model improves the representation of rare nodes, the representation of sparse datasets with few relationships still needs to be enhanced. The experimental results indicate that our model, compared to the base model TransE, shows significant improvements in connection prediction tasks. It indicates that leveraging the attention mechanism can aggregate fixed-size topological neighborhood node information, enhancing the representation of nodes and relations within the knowledge graph. Our model also demonstrated its ability to represent rare nodes. In our next research, we will explore the reasons for this and further optimize our model. One possible direction for optimization is to replace uniform neighborhood

sampling with a non-uniform local neighborhood sampling function. Additionally, we intend to extend the model to include more complex graph networks in the future.

## SOURCE OF THIRD-PARTY DATA

We used the FB15K-237 and WN18RR datasets, and we have referenced and explicated the corresponding original authors in the datasets part.

### Funding
The authors received no funding for this work.

### Competing Interests
The authors declare there are no competing interests.

### Author Contributions
- Ying Liu conceived and designed the experiments, performed the experiments, analyzed the data, performed the computation work, prepared figures and/or tables, authored or reviewed drafts of the article, and approved the final draft.
- Peng Wang conceived and designed the experiments, performed the experiments, analyzed the data, performed the computation work, prepared figures and/or tables, authored or reviewed drafts of the article, and approved the final draft.
- Di Yang conceived and designed the experiments, performed the experiments, analyzed the data, performed the computation work, prepared figures and/or tables, authored or reviewed drafts of the article, and approved the final draft.
- Ningjia Qiu conceived and designed the experiments, performed the experiments, analyzed the data, performed the computation work, prepared figures and/or tables, authored or reviewed drafts of the article, and approved the final draft.

### Data Availability
The FB15K-237 and WN18RR datasets are available at Zenodo: Lzx13567. (2023). Lzx13567/KATKG_rare: KATKG_rare v1.2 (v1.2). Zenodo. https://doi.org/10.5281/zenodo.10228930.

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
