# Peer review of "A knowledge graph embedding model based attention mechanism for enhanced node information integration"

_PeerJ Computer Science, doi:10.7717/peerj-cs.1808_

## Round 0.1 · original submission · Major Revisions

Please take into account the reviewers comments.

**Language Note:** The review process has identified that the English language must be improved. PeerJ can provide language editing services - please contact us at [email protected] for pricing (be sure to provide your manuscript number and title). Alternatively, you should make your own arrangements to improve the language quality and provide details in your response letter. – PeerJ Staff

Reviewer 1 ·

Basic reporting

** Introduction

Not simple enough. Should have a second reading

** Related Work

Acronymous with and without definition.

Experimental design

no comment

Validity of the findings

no comment

Additional comments

no comments

Reviewer 2 ·

Basic reporting

1 The title, “A knowledge graph embedding model with attention mechanism aggregating key information,” is ambiguous. What exactly does “key information” refer to? It's unclear whether “aggregating key information” applies solely to the attention mechanism or the entire model. The authors must rephrase the title for a clearer understanding of the research's objective.

2 The primary concern with this paper is its readability. The manuscript is riddled with grammatical errors and nonsensical sentences, severely undermining the reading experience.

For instance:
(1) In line 22, it reads, “The experiment shows that our model achieves or outperformers baseline model in link prediction, and the embedded representation of rare nodes can be expressed more accurately than the previous model.” Do the authors mean to suggest that their method matches or surpasses the performance of other baseline models? If referencing multiple baseline models, the plural form is appropriate. Also, “the previous model” is ambiguous.
(2) In line 18, the statement “Our model proposes a simple and effective mechanism” is problematic. Can a model truly "propose"? Words like "consists of" or "incorporates" might be more fitting.

(3) In the abstract, the acronym “KATKG” is presented without clarification. The authors must provide the full term when it first appears to ensure clarity for readers.

(4) The authors must include the main experimental results in the abstract, highlighting performance scores and key findings.

(5) The first three paragraphs of the introduction provide excessive background. The authors must swiftly transition to their primary research topic, outlining their objectives and motivations.

(6) The structure and flow of the related work section are wanting. The authors must provide a brief introduction to clarify the organization and also articulate the limitations distinctly.

Experimental design

A comprehensive diagram or visual depicting the method's architecture must be provided at the start of the methodology section.

The authors must present key experimental parameters in a table for clarity. Parameters without academic significance, like the learning rate, should be considered for removal.

Section 4.3.1 seems redundant, given it reiterates models previously discussed in the literature review. Moreover, the authors must think about incorporating recent baseline models like “TransGCN” and “RGHAT”, especially since they're mentioned in the literature review. If these models aren't included, a justification for their exclusion is essential.

Both the baselines and main result sections are erroneously labeled as 4.3.1.

Validity of the findings

The main result section merely lists experimental outcomes, which isn't sufficient. An encompassing discussion, especially highlighting the model's treatment of "rare nodes" and the aggregation of key information, must be included.

Additional comments

The manuscript cannot be considered for publication unless it is thoroughly proofread by fluent English speakers or professional editors.

·

Basic reporting

English language used in the text is on a good level and in professional style.

The abstract and introduction give very good idea of the topic of the research. The cited related literature is relevant. The literature survey is good and positions the work of the authors with comparing it with the existing literature.

"In order to improve search quality and user search experience, Google first proposed the concept of knowledge graph": please cite a publication that proves the statement.

The structure of the publication is with respect to Peerj standards.

References citing must not be in the superscript, but on the baseline of the text like this [1].

Figures. Please restructure your figures, so they will be possible to read. In the presented text the font inside figures is too small.

Tables are relevant and clear. They provide the data to clarify the statements in the text.

Raw data. The links provided for the data sets do not work. Search with Google finds the following links
https://paperswithcode.com/dataset/fb15k-237
https://paperswithcode.com/dataset/wn18rr
Can authors confirm or deny that these are the datasets used in the research?

Can authors share the source code or implementation details for their implementation?

Experimental design

The authors provide description of their model and they describe the training stage of the neural network together with the achieved result. The experiments are conducted on the FB15K-237 and WN18RR datasets.

There are constants that are used by the author, which need more clarification (see line 223). How is the value used chosen?

The experiments described provide good description of the research.

Validity of the findings

The experiments described in the text and tables look statistically well organized.

Additional comments

Generally good proposition. If the authors fix the following details, the text will be a good journal paper to publish:
1) Fix the figures. They must be absolutely clear and readable.
2) Provide details of your implementation. Clarify the used datasets.
3) Clarify how you estimated all constants used in your experiments, even if they are just try-and-error approach.

---

## Round 0.2 · Major Revisions

Please carefully fix the reviewers comments.

Reviewer 2 ·

Basic reporting

The author has made commendable efforts to refine their work. However, there are still notable grammar issues that warrant attention, as they could potentially detract from the overall reader experience.

In line 12, the phrase "in order to applied" appears to be a typographical error. It should be corrected to either "in order to apply" or "in order to be applied."

The sentence "which incorporates graph attention mechanism to integrates key nodes information" should be revised to "which incorporates a graph attention mechanism to integrate key node information."

In the sentence "that our model can embedding rare nodes more accurately than the baseline model," the phrase "can embedding" seems incorrect. It should be revised to "can embed rare nodes."

It is not the reviewer's responsibility to proofread the manuscript, the prevalence of such grammatical errors can impact the clarity and professionalism of the writing. It is suggested that the author employs grammar-checking tools to rectify these issues before the final submission. This not only demonstrates a commitment to presenting a polished manuscript but also respects the time and efforts of both reviewers and readers.

Experimental design

no comment

Validity of the findings

no comment

Additional comments

no comment

·

Basic reporting

### English language

On good and professional level.

### Intro, background and references

Give clear idea about the purpose of the research. References are relevant and correctly cited in the text.

### Text structure

The structure of the text is with respect of the PeerJ standards.

### Figures and tables

Figure 2 can be improved. It is blurry, small and cannot be read. The rest figures are clear.

### Raw data

Authors provided their implementation in Python programming language. Please, do not comment your code in Chinese when you plan to share it for the purpose of English language journal.

Experimental design

### Originality of the research and scope of the journal

The research is original and it is within the scope of the journal.

### Research questions definition

Research questions are clearly and well defined.

### Technical and ethical standards of the research

The technical and ethical standards of the presented work are on high level. Also, the authors clearly cited the datasets they used in their experiments.

### Description detail sufficiency to replicate

The details of the description are on sufficient level.

Validity of the findings

### Impact and novelty

The presented knowledge embedding model is novel and has been verified by the experiments shown in the text.

### Data provided robustness and statistical control

The experiment discussed in the text is statistically meaningful, which can be seen also in the tables.

### Conclusions

Clearly and well formulated.

Additional comments

Please, fix Figure 2. Please, translate the comments in the source code in English.

---

## Round 0.3 · accepted · Accept

Based on the reviewers' comments, the paper can be accepted.

Reviewer 2 ·

Basic reporting

I noticed that the authors have significantly rephrased and improved their writing. While it is not quite native enough, it is overall readable. There are still some minor grammar issues left, for example:

(1) In line 331, the sentence 'the ranges and values of key experimental parameters was shown in Table 3 for clarity' should be 'the ranges and values of key experimental parameters were shown in Table 3 for clarity.'

(2) In line 357, the sentence 'According to statistics, the FB15k-237 dataset contains 36 isolated node' should be 'According to statistics, the FB15k-237 dataset contains 36 isolated nodes.'

I hope the authors carefully proofread the manuscript before the final submission.

Experimental design

no comment

Validity of the findings

no comment

Additional comments

no comment

·

Basic reporting

As in my previous review.

Experimental design

As in my previous review.

Validity of the findings

As in my previous review.

Additional comments

The authors have corrected their work according to my remarks.